# Deep Learning Segmentation and Classification for Urban Village Using a Worldview Satellite Image Based on U-Net

**Zhuokun Pan [1,2,3], Jiashu Xu [4], Yubin Guo [4,*], Yueming Hu [1,3] and Guangxing Wang [2,3]**

[1] College of Natural Resources and Environment, South China Agricultural University, Guangzhou 510642, China; zhuokun.pan@siu.edu (Z.P.); huyueming@scau.edu.cn (Y.H.)

[2] School of Earth Systems and Sustainability, Southern Illinois University, Carbondale, IL 62901, USA; gxwang@siu.edu

[3] Guangzhou Research Institute of Natural Resource Science and Technology, Guangzhou 510640, China

[4] College of Mathematics and Informatics, South China Agricultural University, Guangzhou 510642, China; xujiashu@scau.edu.cn

* Correspondence: guoyubin@scau.edu.cn

**Abstract:** Unplanned urban settlements exist worldwide. The geospatial information of these areas is critical for urban management and reconstruction planning but usually unavailable. Automatically characterizing individual buildings in the unplanned urban village using remote sensing imagery is very challenging due to complex landscapes and high-density settlements. The newly emerging deep learning method provides the potential to characterize individual buildings in a complex urban village. This study proposed an urban village mapping paradigm based on U-net deep learning architecture. The study area is located in Guangzhou City, China. The Worldview satellite image with eight pan-sharpened bands at a 0.5-m spatial resolution and building boundary vector file were used as research purposes. There are ten sites of the urban villages included in this scene of the Worldview image. The deep neural network model was trained and tested based on the selected six and four sites of the urban village, respectively. Models for building segmentation and classification were both trained and tested. The results indicated that the U-net model reached overall accuracy over 86% for building segmentation and over 83% for the classification. The $F_1$-score ranged from 0.9 to 0.98 for the segmentation, and from 0.63 to 0.88 for the classification. The Interaction over Union reached over 90% for the segmentation and 86% for the classification. The superiority of the deep learning method has been demonstrated through comparison with Random Forest and object-based image analysis. This study fully showed the feasibility, efficiency, and potential of the deep learning in delineating individual buildings in the high-density urban village. More importantly, this study implied that through deep learning methods, mapping unplanned urban settlements could further characterize individual buildings with considerable accuracy.

**Keywords:** deep learning; urban village settlement; Worldview imagery; U-net; segmentation; Guangzhou

## 1. Introduction

In less developed countries like Asian and Africa, urban sprawling is usually accompanied by the emergence of unplanned settlements, such as slums, shantytowns, and urban villages [1,2]. Rapid urbanization with the housing demand of low-income city dwellers has contributed to the emergence of unplanned settlements [3–5]. This kind of built-up area is often stuffed with high-density small buildings. Although unplanned settlements do provide housing for low-income city dwellers, their existence contributes to unequal living conditions, inhabitants often do not have access to equal services

and face unsanitary living conditions, and there are issues with public safety [4,6]. Effective management, improvement, and reconstruction of the unplanned settlements become important policies in some cities [2,3]. Identifying and characterizing the unplanned settlements is essential and indispensable for urban planners and policymakers to evaluate urban reconstructions [1,2,4,7]. Typically, the urban village is the most commonly existing unplanned urban settlement in China.

Traditionally, the cartography of the unplanned buildings in urbanized areas has relied on field surveys by land management departments through the collection of measurements of buildings and digitalization [1]. However, field surveys are costly and labor-intensive due to the extensive existence of urban villages [8]. Remote sensing technologies have the advantage of being low-cost and allowing large coverage; without a doubt, it is an add-on method to the field surveys, and helps update the base map data [1,9]. Using low-altitude aerial photos or Unmanned Aerial Vehicle images to capture the buildings and digitize their footprints, the manual method is another conventional mapping method, but it is time-consuming. Therefore, a highly efficient, intelligent, and image-based methodology is urgently needed to cope with the challenges.

Remote sensing-based mapping of urban buildings has a long history with substantial literature [10]. With the increasing availability of high-resolution optical images, along with object-based image analysis (OBIA) has dominated the area of mapping urban buildings [9,11–13]. More often, most current techniques for image target detection are based on spectral and spatial features [14]. In recent years, Random Forest (RF) as a machine learning method has become one of the most popular approaches in built-up areas mapping [15,16]. Although numerous classification methods have been developed for urban land use mapping for some high-density built-up areas, the widely used pixel- or object-based methods usually does not allow the extraction of individual buildings, but typically clusters several buildings into one segment [17]. In a complex urban built-up area, the features (spectral, texture, shape, etc.) are uneasily described by conventional remote sensing methods, owing to the large variance among urban buildings [7,18–20]. Segmentation is the process of partitioning an image into segments by grouping neighboring pixels with similar feature values (brightness, texture, color, etc.) [13]. In high-density built-up areas, segmentation with OBIA is often impeded by difficulties such as the scale selection and rule definition. It is challenging to completely delineate the boundaries and preserve their shapes because the noise and textures on the building's edges usually degrade the performance of image segmentation [21]. Additionally, a generalized and straightforward methodology is often hard to obtain and has not been reported. Therefore, developing a reliable and accurate building segmentation method towards mapping the unplanned urban settlements is still challenging.

On the other hand, there have been several studies demonstrating that the high-density slums can be mapped out using remote sensing images based on their physical characteristics distinguishable from formal settlements [1,2,4,16]. Meanwhile, it is crucial to recognize that mapping the slums need to go beyond delineating the whole area, characterizing individual buildings accounts for more demands. For instance, to assess the potential and benefits of urban reconstruction, both governmental and private decision-makers need to be acquainted with information regarding reconstruction incentives, public services, and environmental improvements [22,23]. The valuable information can only be obtained based on individual buildings extracted to address the need of different stakeholders. Therefore, the classification and characterization of buildings appear equally important.

Semantic segmentation is a classic Computer Vision problem to mask out regions of interest. In recent years, machine learning technologies, especially deep learning, have attracted the attention of the remote sensing community [14,17,24–27]. The deep convolutional neural network (CNN) has been applied to semantic segmentation [20,27]. One of the most recognized deep learning algorithms for image segmentation is the fully convolutional network illustrated by Long et al. [28]. This deep learning architecture is an end-to-end, pixel-by-pixel manner of semantic segmentation. It captures the details of features, and transfers the details to be recognizable in the trained neural network. Therefore, CNN is gaining more attention, attributing to its capability to automatically discover relevant contextual features in image categorization [28,29]. The CNN-based semantic segmentation has been applied

to many pixel-wise image applications, such as road extraction, building extraction, urban land use classification, maritime semantic labeling, vehicle extraction, damage mapping and so on [1,24,30,31].

Conventional image classification methods have encountered bottlenecks for high-density built-up areas. CNN often outperforms most conventional image classifications [1]. Potentially, it could segment individual buildings in high-density unplanned settlements. Deep learning-based semantic segmentation is to use a convolutional neural network to learn prior knowledge of features and extract the objects from images. There are several advantages of using CNN to find solutions for unplanned urban settlements. First of all, it is fast to apply CNN to extraction. The learning process does not require human interventions. The network training takes hours on a computer. After the network training is completed, the classification of each image takes only seconds [26,32]. Secondly, it is truly automatic. The intrinsic difference between deep learning methods and the traditional visual recognition methods is that the deep learning methods can autonomously learn feature representations from a large amount of data, without much expertise or effort in predefining features [7,32]. In a complex urban area, manually predefined features are not likely able to cover all land cover types. Therefore, it is of great significance for automatic feature learning from remote sensing imagery rather than manually-defined features [26]. Moreover, the advantage of deep learning for a remote sensing image is characterized by sample repeatability and adaptability. Training samples are reusable in the CNN, which is significantly minimizing manual efforts in hand-craft features. More importantly, for most conventional remote sensing methods, boundary segmentation between high-density buildings is very difficult to achieve. This bottleneck could be potentially overcome by deep-learning semantic segmentation [19,26,27].

In recent years, many CNN with excellent performance has been reported. Compared to others, the U-net proposed by Ronneberger et al. [33] appears to be more popular and more quickly adopted and modified for remote sensing image segmentation. Training with a substantial amount of training data, the U-net uses a sliding-window set up to predict the class label of each pixel by providing a local region (patch) around that pixel. It also works with notably few training samples and yields precise semantic segmentation. The U-net has been adopted and highly evaluated by researchers [19,20,30,31,34].

Although remote sensing detection has greatly benefited from deep learning methods, whether the deep learning methods can separate the adjacent buildings in an overcrowded urban village has not yet been understood. Semantic segmentation is naturally related to classification problems. However, the feasibility, capability, and accuracy of the U-net convolutional neural network for classification in high-density urban village buildings had not been well understood either. Thus, the research objective of this study was placed on the semantic segmentation and classification for high-density buildings in urban villages. In this article, a deep learning framework based on U-net for the semantic segmentation of high-resolution images was proposed. By implementing and applying the U-net based CNN, the mapping capability of the individual buildings in the urban villages was demonstrated, and accuracy was validated.

## 2. Study Area and Data

### 2.1. Study Area

This study focused on building segmentation in the high-density urban villages located in Tianhe District of Guangzhou City, Southern China (Figure 1a), where a fast land-use conversion during its urbanization has been witnessed [35]. The urbanization process has led to the emergence of skyscrapers. However, lots of urban villages were submerged in the modern city. The urban villages as unplanned settlements typically exist in all large cities of China [7]. Thus, characterizing the unplanned settlements is necessary for urban management, livelihood improvement, and reconstruction.

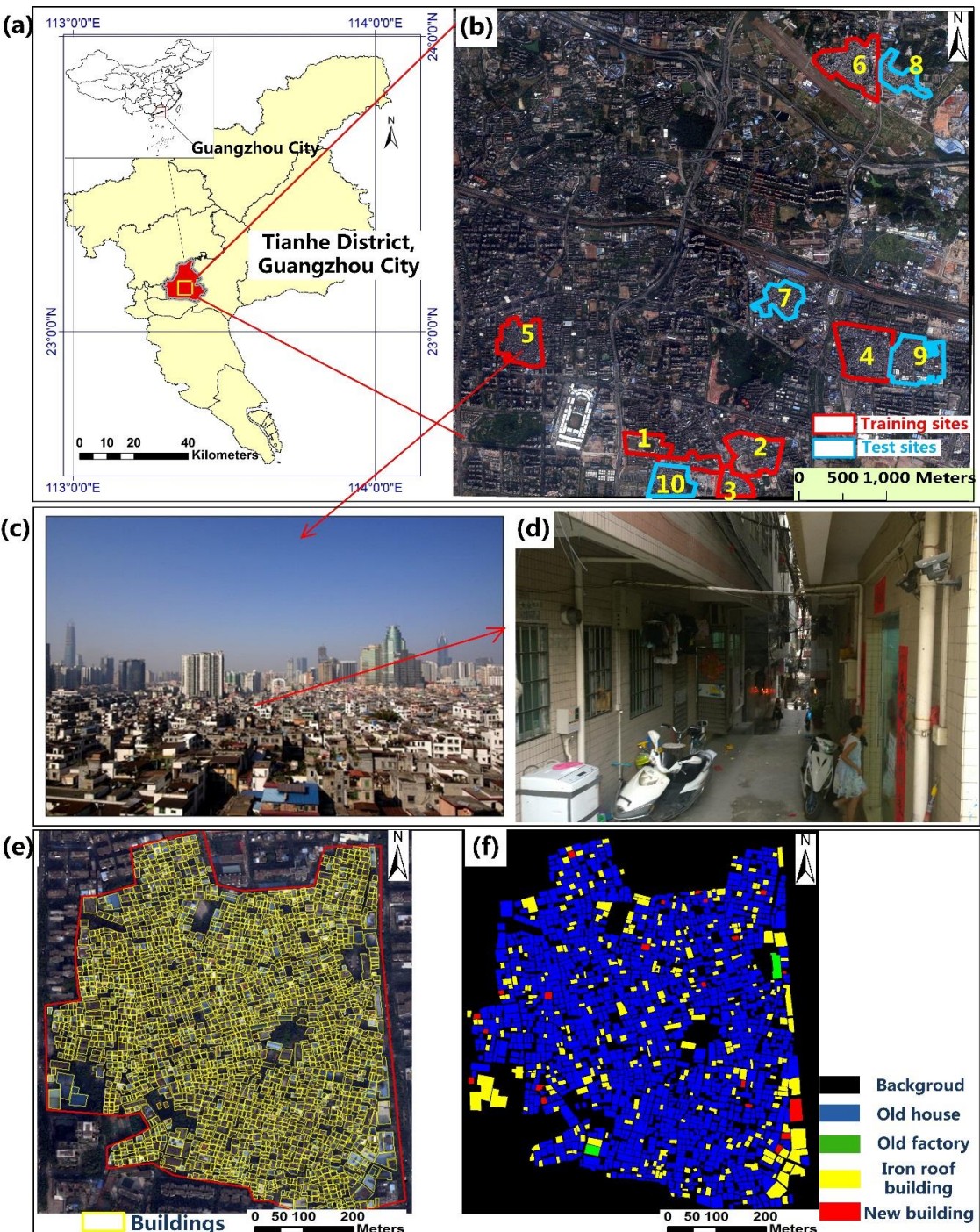

**Figure 1.** (**a**) Location of the study area in Tianhe District of Guangzhou City; (**b**) WorldView-2 image with red and blue polygons marked as training and test sites, respectively; (**c**,**d**) on-site view of one selected site; (**e**) vector file of the building boundaries; and (**f**) labeled image with building categories.

Since 2009, Guangzhou City has begun its urban redevelopment so-called "Three Old Transformation" which was mainly dedicated to eliminating and reconstrucingt urban villages. The urban villages (Figure 1b–d) have been formed by the soaring demand for low-cost residential homes since the 1980s and still serve a housing supply for city immigrants from rural areas. There are unsanitary conditions, limited open space and greenery, and poorly managed facilities in the urban villages [7,36], which do not seem to fit the requirements of the formal urban planning. The potential risk of safety and criminals are often public concerns inside the urban villages [36,37]. The redevelopment

of the urban villages has gradually become significant and necessary in Guangzhou City [37,38]. The city government of Guangzhou, such as Guangzhou Urban Renewal Bureau, has a strong need to identify the old buildings and obtain their detailed information to carry out the reconstruction of the urban villages.

## *2.2. Datasets*

A high spatial resolution Worldview-2 satellite image acquired in December 2013 was used, and the scene of the image covered the central part of Tianhe District (Figure 1b). The Worldview-2 satellite was launched in October 2009 and is the first high-resolution satellite with 8-multispectral imaging bands. The image has eight 2-m resolution multispectral bands and a 0.5-m resolution panchromatic band. Since the Worldview-2 satellite is capable of stereo imaging, the nadir view image was considered appropriate for urban mapping [39]. The atmospheric correction was applied to the image to obtain the values of spectral reflectance. The multispectral bands were pan-sharpened to a spatial resolution of 0.5 m × 0.5 m using the nearest neighbor diffusion (NNDiffuse) pan sharpening algorithm [40], through which the spectral and texture details were fully preserved.

There were ten urban villages with overcrowded buildings existing in this scene of the image. Six of the sites marked with red were selected as training data, while the remaining four sites marked with blue were used as test data (Figure 1b). A building boundary vector file (*.shp*) of the urban villages was obtained courtesy of Guangzhou Land Resources and Urban Planning Bureau. The building boundary file and the satellite image had a strict alignment in spatial registration. The vector file was converted to a label image to train the U-net model (Figure 1e). The building boundaries were associated with their attributes, which allowed classifying the buildings as categories: "Old house", "Old factory", "Iron roof building", and "New building" (Figure 1f). The satellite image and the building vector file was preprocessed in *ArcGIS 10.2* and *ENVI 5.3*. The labeled image was generated from the building vector file and were converted to the classes.

## 3. Deep Learning Segmentation Using U-Net

Semantic segmentation is a process of taking an image and labeling each pixel in that image with a specific class [27,33]. More often, these processes are completed by a deep convolutional neural network (CNN). Through assistance from Computer Vision, semantic segmentation segments the image automatically. To do that, researchers can either use a pre-trained CNN to perform segmentation, or create a CNN; by labeling the targeted objects, researchers use the labeled data to perform network model training. In this study, the procedure was summarized as follows: (1) A large number of building footprints from the labeled image were converted from a building vector file; (2) The training data was partitioned into training sets and validation sets, then the satellite image associated with the label data was inputted to train a network model, and generate a model training report; (3) The model was applied to given test data. After the model training was completed, the test data was input into the network model, and the model was able to segment the targets from the background; and (4) The results were evaluated.

### *3.1. Network Architecture*

In this study, semantic segmentation was developed based on U-net architecture [33]. It is noteworthy that an encoder-decoder architecture becomes increasingly popular in semantic segmentation due to its high flexibility and performance [28,30,34]. According to Ronneberger et al. [33], the U-net architecture uses the following types of layers and special operations: (1) Conv2D, Simple convolution layers with padding and 3 × 3 kernel; (2) MaxPooling2D, Simple max-pooling layers with 2 × 2 kernel; (3) A cropping 2D, cropping layer used to crop feature maps and concatenate; (4) A concatenate layer used to concatenate multiple feature maps from different stages of training; (5) An UpSampling2D layer used to increase the size of the feature map. Then, the decoder part is implemented for up-sampling; and (6) Finally, a softmax layer is added to generate a final segmentation

map. The structures of the encoder and decoder parts are symmetrical with skip connections between them, which proves to be effective to produce fine-grained segmentation results. More importantly, U-net can preserve the feature maps to the same size as the original image, where up-sampling layers are followed by several Conv layers to produce dense features with finer resolutions. This study modified the deep-learning architecture to handle the Worldview image with eight bands. The modified deep learning architecture can accept any size of images ranging from megabyte to gigabyte. The advantage of the modified U-net is an "end-to-end" procedure; for instance, segmentation. For high-density building segmentation using images, the modified method can assign a class to every pixel according to which class exactly belongs to the intricate.

Figure 2 gives an intuitive demonstration of the U-net structure. Like other commercial networks, the architecture of the U-net consists of a large number of different operations illustrated by small arrows. There are several details and basic concepts that are applied to the U-net: (1) the convolutional layers for feature extraction through multiple 3 × 3 convolution kernels (denoted by Convolution); (2) the batch normalization layer for accelerating convergence during the training (denoted by Batch Normalization) [41]; (3) the activation function layer for nonlinear transformation of the feature maps, in which we adopted the widely used rectified linear unit (ReLU) (denoted by Activation) [42]; (4) the max-pooling layer for down-sampling of the feature maps (denoted by Max-pooling) [43]; (5) the up-sampling layer for recovering the size of the feature maps that are down-sampled by the max-pooling layer (denoted by up-sampling); and (6) the concatenation layer for combining the up-sampled feature map in the deep layers with the corresponding feature map from the shallow layers (denoted by Concatenation).

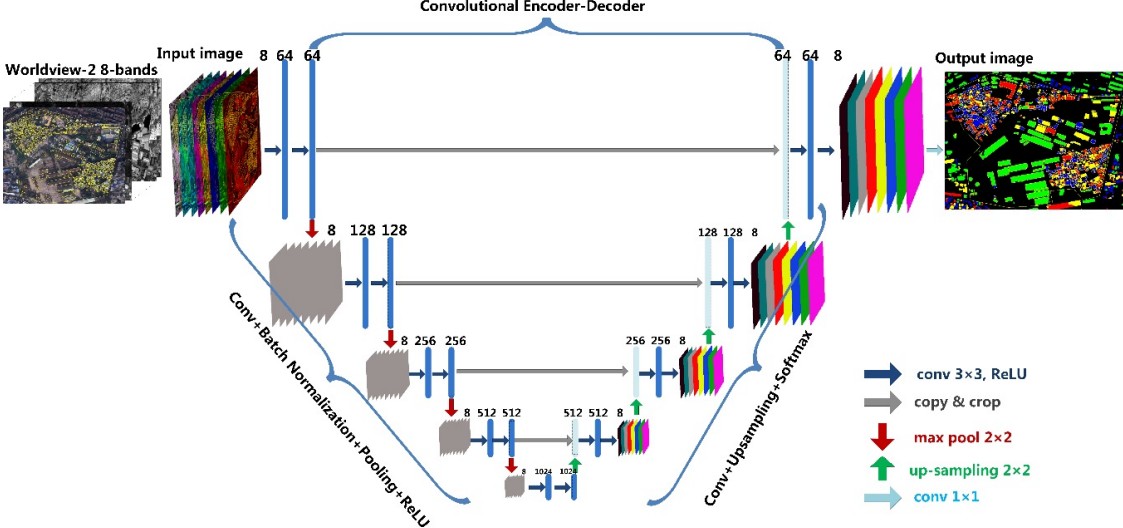

**Figure 2.** Illustration of the U-net Architecture for semantic segmentation, including the name and function of each layer (modified from Ronneberger et al. [33]).

In brief, the U-net is characterized by the convolutional max-pooling, cropping, concatenate, and up-sampling layers. It receives the input image and runs through two convolutional operations with ReLU activation. The image is then encoded into the pooling layer. This process happens a few times, reducing the size of the feature maps. Once 1024 sample maps are obtained, the model starts up-sampling. The layers of the feature maps are concatenated with the feature maps from the down-sampling process. The feature maps are up-sampled from the previous concatenated feature maps and the model finally outputs the segmentation map as the same size of the original image. From Figure 2, the data are propagated through the network by convolutional encoding and decoding with copy and crop steps to retain the image information. Along all the possible paths, at the end, the segmentation map is obtained.

*3.2. Network Model Training*

The labeled image and 8-band Worldview image were used to train the deep CNN model. In this study, the tiles with the size of 256 × 256 pixels were cropped from the image. The randomly cropped tiles were the candidate training datasets. The more the training data are added, the more robust the network training, the better the segmentation results. The data augmentations were then applied on the tiles by rotating, mirroring, brightness enhancement and adding noise points to increase the features of data. In the experiments, 30 epochs with 16 batches per epoch were applied, and the learning rate was set at 0.01 for model training.

From the training dataset (Figure 1b, the red boundary areas), the buildings in the training sites were randomly divided into training samples of 70% and validation samples of 30% for model training, respectively. The epoch's value was used to maintain the accuracy and convergence of loss. It would determine how well the model would perform and also influence the network training time. In this experiment, the Stochastic Gradient Descent optimization algorithm was adopted in the optimizer [44]. The learning rate in the optimization determines the speed of learning process, making the network training to converge. Weight decay is a penalty added to the loss function to prevent over-fitting of the network model, and the used value was 0.0001. Momentum means to what extent the model remains the original updating direction, and the value range was set up as 0.9. In this experiment, the following cross-entropy as a loss function was adopted to evaluate the training performance.

$$L_{loss} = -\frac{1}{N}\sum_{i=1}^{N} y_i log\hat{y}_i - (1 - y_i)log(1 - \hat{y}_i) \tag{1}$$

where $y_i$ is the $i$th actual labeled category and $\hat{y}_i$ is the $i$th predicted category.

Figure 3 presents the model training process. As the number of epochs increased, the accuracy of the training and validation increased, while the loss decreased. The accuracy increase and loss decrease changed rapidly at the beginning, then slowly and gradually became stable after 15 epochs. The validation loss value started to become stable after 0.15. We found that the network model obtained from the training sites could reach an accuracy of over 90% to identify the buildings in the training areas. The model training took about 72 hours to finish with 30 epochs and 10,000 cropped sample images.

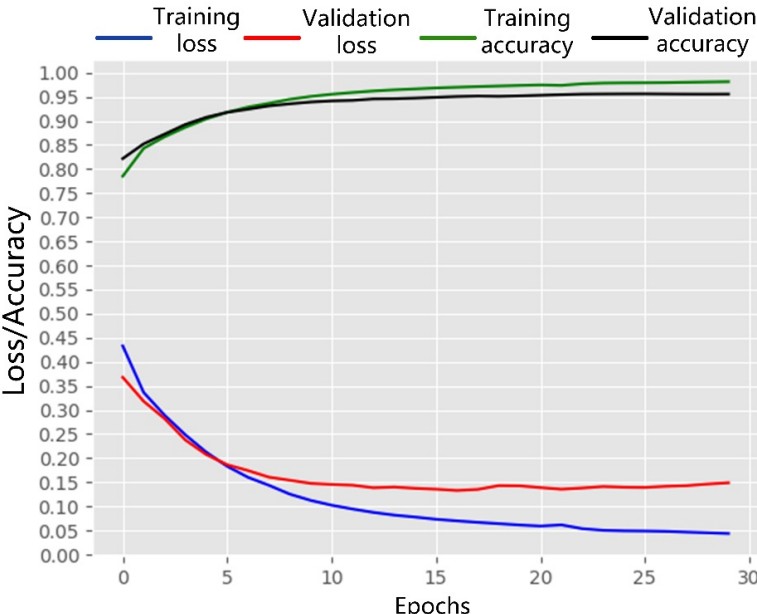

**Figure 3.** Training loss and accuracy against 30 epochs: training loss (blue); validation loss (red); training accuracy (green); validation accuracy (black).

### 3.3. Experiment Environment and Programming

The U-net architecture as well as the whole semantic segmentation procedure were mainly implemented based on the *Tensorflow* and *Keras* deep learning framework in *Python* programming language. All other processing and analyses were carried out using open-source modules, including *GDAL*, *NumPy*, *Pandas*, *OpenCV*, *Scikit-learn*, etc. The deep learning network experimentation and modeling were executed in *Jupyter Notebook* programming platform. The configuration of the computer environment was *Ubuntu* 16.04.4 operation system with the virtual machine configured with 16 GB of RAM and an *NVIDIA Geforce GTX 1080* GPU. The complete procedure of the deep-learning for semantic segmentation using the U-net was summarized in Figure 4.

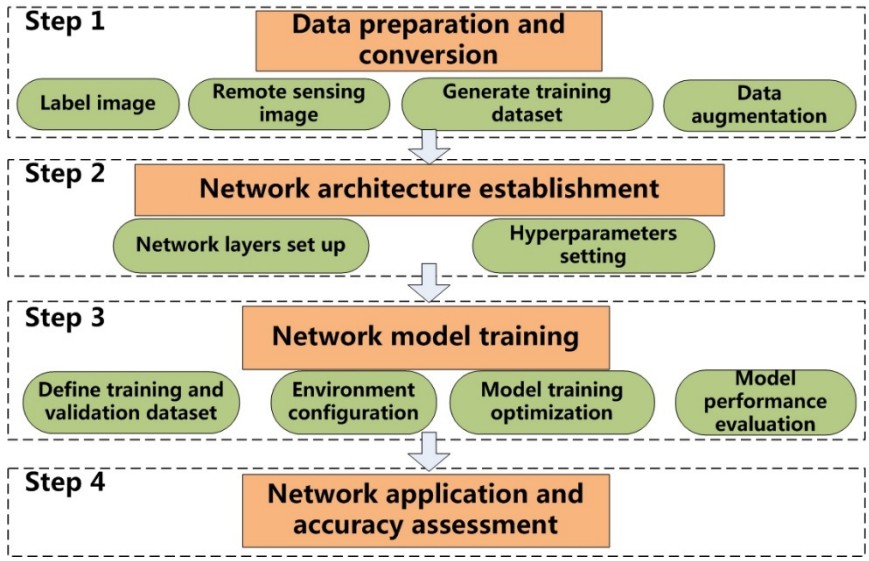

**Figure 4.** Procedure of the deep neural network training and application.

### 3.4. Accuracy Assessment

In this study, the performance of semantic segmentation using U-net was evaluated by the confusion matrix [45]. The overall accuracy (*OA*) was calculated by summing the percentages of pixels that were correctly recognized by the model compared to the reference labeled image for all the categories (Equation (2)).

$$OA = \frac{\sum_{i=0}^{k} p_{ii}}{\sum_{i=0}^{k} \sum_{j=0}^{k} p_{ij}} \tag{2}$$

where $p_{ii}$ means the number of pixels for categories $i$ correctly recognized by the model, while $p_{ij}$ means the number of pixels for categories $i$ incorrectly recognized into category $j$ by the model, and $k$ is the number of categories.

The confusion matrix makes it easy to compute and apply $F_1$-score to evaluate the semantic segmentation, which is based on the Precisions and the Recalls (Equation (3)). Table 1 gives an example of the confusion matrix for building extraction. The columns represent the actual values, and rows represent the predicted values. The confusion matrix is a way of classifying true positives (*TP*), true negatives (*TN*), false positives (*FP*), and false negatives (*FN*) when there are more than two classes. Then, $F_1$-score (Equation (4)) is then computed to evaluate the performance of a model that considers a

balance between Precision and Recall. For segmentation and multi-class classification, the calculation is similar, and an average of the $F_1$-score values of all the classes can be obtained.

$$Recall = \frac{TP}{TP + FN}, Precision = \frac{TP}{TP + FP} \tag{3}$$

**Table 1.** $F_1$-score calculation from confusion matrix.

|  |  | Ground Truth | |
| --- | --- | --- | --- |
|  |  | Buildings | Non Buildings |
| **Prediction** | Buildings | TP | FP |
|  | Non Buildings | FN | TN |

The Recall is the ratio of the correctly predicted positive observations to all observations in an actual class. The Precision is the ratio of the correctly predicted positive observations to the total predicted positive observations. The $F_1$-score combines the Precisions and the Recalls.

$$F_1 - score = 2 * \frac{Precision * Recall}{Precision + Recall} \tag{4}$$

The Jaccard Index, also known as Intersection-over-Union (IoU), which means the ratio of intersection and union of two sets, is a statistic used in understanding the similarities between sample sets. In image semantic segmentation, these two sets represent the prediction and the reference. When a segmentation image is obtained, the value of IoU for each category will be calculated according to Equation (5).

$$IoU = \frac{|A \cap B|}{|A \cup B|} \tag{5}$$

where A represents the prediction and B represents the ground truth.

To evaluate the performance of the deep learning for semantic segmentation using U-net, we presented the intuitive visualization of the predicted results compared with the ground truth, calculated the overall accuracy, $F_1$-score, and Intersection-over-Union value. Additionally, the performance of U-net was compared with those from Random Forest (RF) and object-based image analysis (OBIA).

## 4. Results

### 4.1. Performance of Building Segmentation

The results of building segmentation were presented in Figure 5, including the six training sites and four test sites within the red box and blue box, respectively. The predicted images in the training sites and test sites showed a good match with the label image (ground truth). For each high-density urban village with overcrowded, irregular-shaped buildings, the network model precisely delineated individual buildings. As presented in Figure 5, the data set used for training should get good prediction results; on the other hand, the test sites that also obtained good predicted maps visually looked like the label images. A snapshot of a magnified area also gave an in-depth inspection of the segmentation results; more importantly, it is shown that the boundaries of the adjacent buildings were correctly separated. Individual buildings are seldom missing, and boundary shapes are basically delineated and preserved.

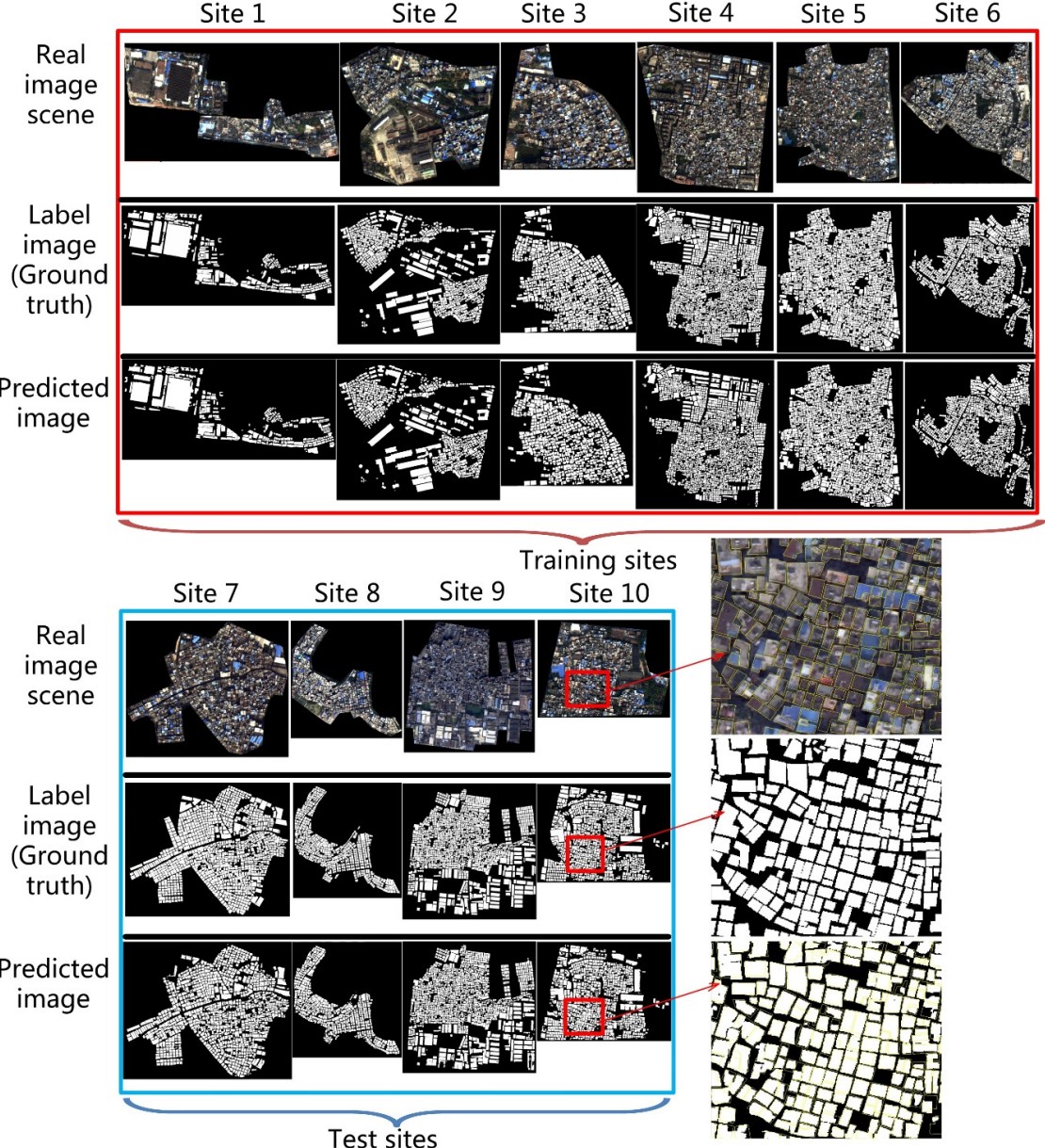

**Figure 5.** Performance of building segmentation by U-net.

The overall accuracy, $F_1$-score and Intersection-over-Union were summarized in Table 2, to give a quantitative description of the prediction results. All the training sites achieved overall accuracies of over 93%, and the test sites achieved an overall accuracy of over 86%. Moreover, both the training and test sites led to $F_1$-score values over 0.89, and an Intersection over Union value over 90%, both of which indicate the excellent performance of the building's segmentation.

**Table 2.** Accuracy assessment of building segmentation.

|  | **Training Sites** | | | | | | **Test Sites** | | | |
|---|---|---|---|---|---|---|---|---|---|---|
|  | **1** | **2** | **3** | **4** | **5** | **6** | **7** | **8** | **9** | **10** |
| Overall accuracy (%) | 98.97 | 97.59 | 96.03 | 93.28 | 95.15 | 97.24 | 87.70 | 95.17 | 96.92 | 86.88 |
| $F_1$-score | 0.993 | 0.984 | 0.972 | 0.895 | 0.957 | 0.982 | 0.909 | 0.971 | 0.976 | 0.898 |
| IoU (%) | 92.31 | 90.61 | 96.60 | 95.88 | 99.44 | 88.49 | 93.04 | 99.46 | 98.43 | 97.01 |

### 4.2. Performance of Building Classification

To further demonstrate the capability of the U-net in building classification for the urban village, we trained and applied the network model based on the categories of the buildings ("Old house", "Old factory", "Iron roof building", "New building"). The building classification was presented in Figure 6, including the six training sites and four test sites within the red box and blue box, respectively. As presented in Figure 6, the data set used for training were supposed to get good prediction results; on the other hand, the test sites also obtained good predicted maps that visually looked like the label images. It was found that there were perfect matches between the predicted images and the labeled maps. The adjacent buildings were successfully separated; building types were classified with a few misclassifications. The snapshot of a magnified area further provided a detailed inspection of the results. The shape of a buildings' boundary is basically delineated and preserved.

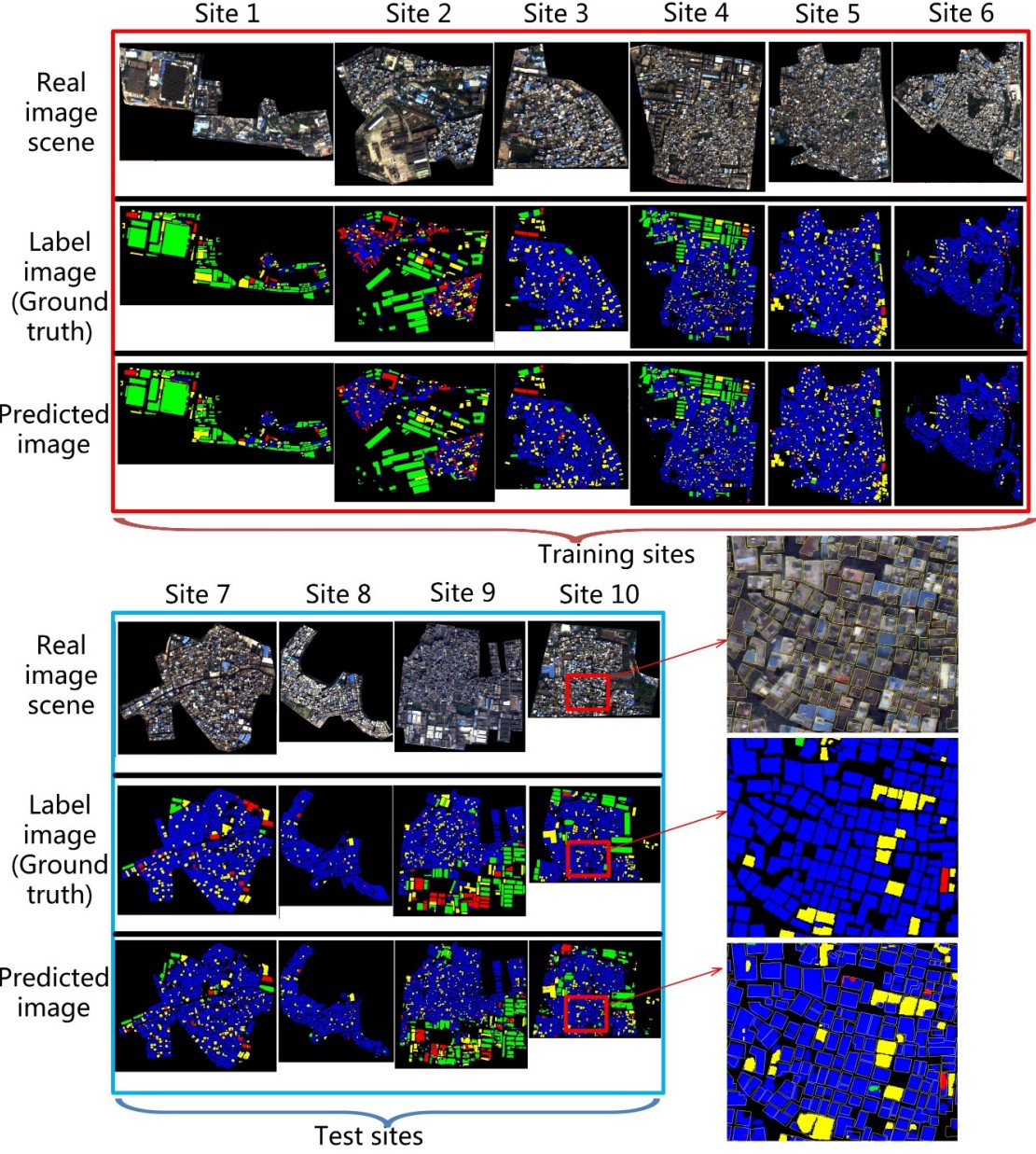

**Figure 6.** Performance of building classification by U-net (blue: Old house; green: Old factory; yellow: Iron roof building; red: New building).

Table 3 listed the summaries of the overall accuracy, $F_1$-score, and Intersection-over-Union to give a quantitative description of the prediction results. As expected, the training sites reached a satisfactory result for all the indicators; the test sites obtained the overall accuracy over 83%, $F_1$-score varied from 0.63 to 0.88, and the Intersection-over-Union value varied from 86% to 93%. Compared with those from the training sites, the results of the test sites were unsatisfactory, but the performance of building classification was reasonable considering the complex urban landscapes.

**Table 3.** Summary of the overall accuracy of building classification.

| | Training Sites | | | | | | Test Sites | | | |
|---|---|---|---|---|---|---|---|---|---|---|
| | **1** | **2** | **3** | **4** | **5** | **6** | **7** | **8** | **9** | **10** |
| Overall accuracy (%) | 96.5 | 93.6 | 93.7 | 94.0 | 94.5 | 93.7 | 83.4 | 87.2 | 86.4 | 84.7 |
| $F_1$-score | 0.96 | 0.94 | 0.93 | 0.92 | 0.89 | 0.89 | 0.68 | 0.88 | 0.65 | 0.63 |
| IoU (%) | 85.52 | 98.40 | 96.11 | 92.88 | 91.87 | 96.00 | 86.20 | 93.26 | 88.78 | 86.11 |

*4.3. Comparison with Random Forest and Object-Based Image Analysis*

The performance of the U-net segmentation was compared with RF and OBIA using test data Site 7. RF was implemented with *EnMAP-Box 2.1* [46], and OBIA was implemented with Rule-based Feature Extraction in *ENVI 5.3*. The results in Figure 7 showed that the U-net obtained the highest overall accuracy of classifying the urban village buildings, followed by RF. OBIA had an inferior performance.

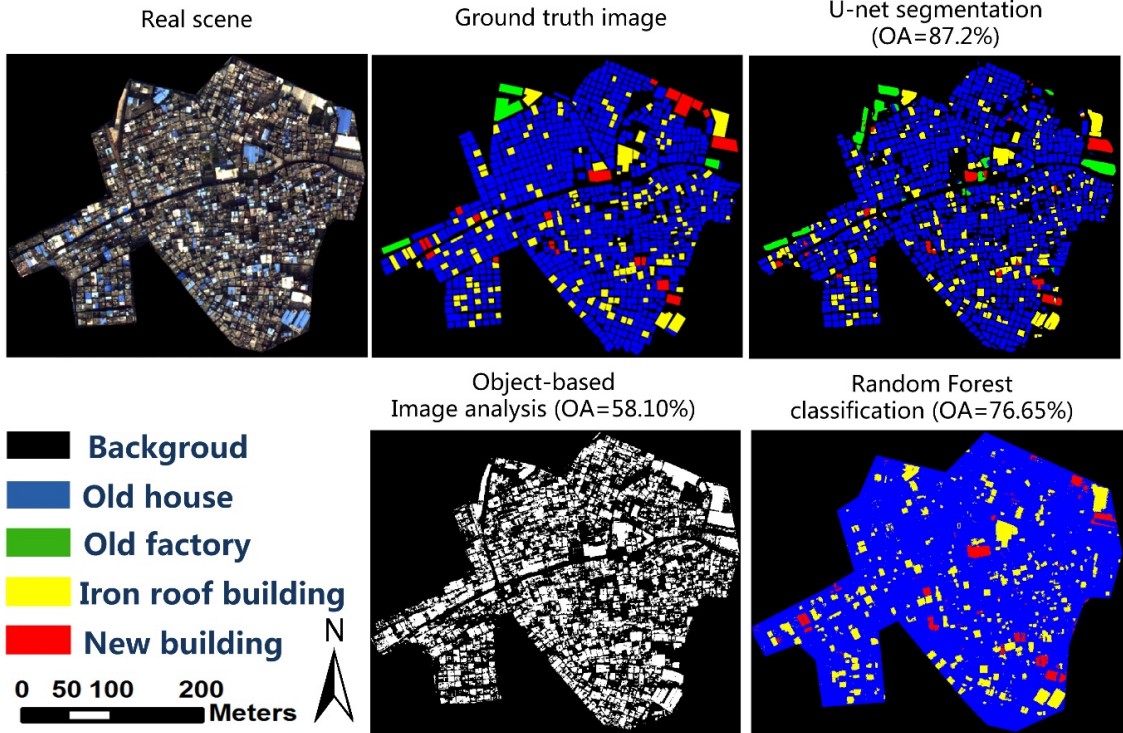

**Figure 7.** Comparison of the building extraction among the U-net, object-based image analysis, and Random Forest.

## 5. Discussion

A lot of artificial intelligence architectures have been developed for feature extraction in remote sensing. Deep learning methods are relatively new and provide great potential for object extraction from remote sensing images. This article presented a deep learning application using U-net for the segmentation and classification of individual buildings in the high-density urban villages located in

Guangzhou City. The experiment achieved reasonable, accurate, and encouraging results, and provided valuable implications for those who are studying the unplanned urban areas.

## 5.1. Result Interpretation

The network model performed well in segmenting the high-density buildings. The results in Figures 5 and 6 showed that most individual buildings were accurately separated, which demonstrates the ability of the Worldview satellite image and the U-net to delineate such kind of buildings in the high-density urban villages. Without a doubt, the U-net segmentation (OA = 87.2%) substantially outperformed the RF (OA = 77.7%) and OBIA (58.1%) methods (Figure 7), and offered valuable implications for large-scale building extraction of complex urban environments.

In addition to the classification accuracy, the convenience of the methods also matters. The deep learning-based semantic segmentation can tackle the high-density urban villages and capture the complicated semantics but requires fewer manual interventions during the process. The OBIA requires user's interactions for the feature design and scale selection but would not ensure a higher detection accuracy. As one of the most favorable machine-learning methods, RF is easy to implement, but the accuracy does not seem an effective application. The RF also failed to separate the adjacent buildings. Thus, the complex urban village environment makes it challenging to use conventional image classification methods to separate the adjacent buildings. Fortunately, deep learning provides significant potential in this regard.

## 5.2. Relevant Studies and Limitations

Detecting and mapping unplanned settlements (e.g., urban slums) had for a long time been considered challenging using remote sensing methods [2]. The previously recognized work of detecting slums and unplanned settlements have mainly focusing on spatial detection and delineation of outer boundaries [47–49]. Since unplanned settlement is stuffed with high-density buildings, studies using the conventional remote sensing methods to delineate the individual buildings are limited. Although deep convolutional neural networks had already been adopted by some authors as well as proposing modified networks, however, their studies focus only on the whole built-up areas and not on the separation of the individual buildings [1,7,50]. Additionally, it is important to realize that mapping the unplanned urban settlements needs to go beyond delineation of the building boundaries and should also characterize the individual buildings, i.e., classify the buildings into different categories, such as "Old house", "Old factory", "Iron roof building" and "New building" in this study. Hence, this study has made contributions to this field.

On the other hand, although the open-access training datasets provide the image and reference datasets for the deep learning network training and testing. However, the on-line datasets were selected for the typical case but not generalized towards the environment of complex unplanned urban settlements [18,49]. This study took the realistic demand into account and investigated the possibility, plausibility, and accuracy of using the deep learning method in a real application. Whether the U-net based segmentation is applicable for those extremely irregular-shaped and high-density urban areas, is the primary question to answer in this study.

## 5.3. Model Improvement

There are still several issues to discuss regarding the efficiency of the network model training. The potential and significance of the deep learning method to extract objects are relying on its fast and full automation, sample repeatability, and adaptability. The performance of the CNN-based image semantic segmentation is dependent on a large amount of training data. A model that works with notably fewer training samples and yields accurate results is highly desirable [24–26]. It is still hard to tell how many samples are sufficient to train a robust network model. Therefore, conducting the sensitivity analysis regarding the number of training samples and the model hyperparameters on

the network performance is necessary for the future. Besides, there is also some potential for further improvement related to the design of the CNN, data augmentation, finer resolution of imagery, etc.

The images used in most articles mainly consist of three channels (Red, Green, and Blue). The Worldview image used in this study has eight bands, which may be beneficial in characterizing the individual buildings because, by involving more bands, the convolution filtering can enhance the extraction of texture, shape, and edge features of the buildings. This perspective supports the proposal that a multi-spectral image outperforms three channels image for the deep neural network training [51]. Further research should also consider the use of auxiliary geo-information such as digital surface models and open-street maps to enhance the model training [2].

*5.4. Lesson Learned*

Remote sensing technologies would not be successful without the support of other theories such as physics, statistics, data fusion, and machine learning [24,26]. This study demonstrated that deep learning could solve the real problem of characterizing the buildings in the complex urban villages, while the widely-used classification methods such as RF and OBIA failed to achieve. When using the deep learning method for the remote sensing applications. However, it is not enough to know how to execute the computer program to run the model; it is also encouraged to have a more in-depth understanding of the object features, and introduce the properties into the deep neural network models for improvement.

We realized that one of the critical factors that affect the performance of deep learning applications in remote sensing is the quality and authenticity of data. Only reliably labeled data of the building features can lead to a robust CNN model. In this study, the features of the buildings in the reference dataset might be inconsistent with those shown in the real image. For example, some buildings show iron roofs in the image but are assigned with the attribute of old houses in the reference dataset. The faults of the experiment data might have been responsible for some of the poor results.

On the other hand, the classifications of the test sites 7, 9, and 10 gave relatively lower values of $F_1$-score, mainly because some newly-built buildings (red in Figure 6) were incorrectly identified. In fact, it is hard to discriminate between the new and old buildings with the naked eye, and the network model learned such features insufficiently, which led to misclassification. The quality of the satellite image also matters. The highly irregular-shaped and great unbalance of the buildings among the samples used for the model training might also have led to problematic results. Last but not least, the labeled reference data and the satellite image should have a strict alignment in spatial registration. Therefore, the researchers should be cautious regarding data quality.

## 6. Conclusions

Unplanned urban settlements widely exist in less-developed countries. Automatic building extraction in the urban environment has long been challenging due to the high-density, complex surroundings, and various shapes of the buildings. Accordingly, the motivation of this study was to show the capability of deep learning for the urban village mapping in which both segmentation and classification of individual buildings were tested. A deep learning paradigm based on U-net CNN was proposed for the semantic segmentation using a Worldview satellite image. The results indicated that most adjacent buildings were well separated; boundary shapes are basically delineated and preserved. The proposed method led to overall accuracy over 86% for the building segmentation, and over 83% for the building classification; $F_1$-score and Interaction-over-Union value both indicated satisfactory results. The deep learning based on U-net significantly outperformed the widely used RF and OBIA, indicating that the deep learning provided greater potential to characterize the individual buildings in the complex urban villages. This study demonstrated the feasibility, efficiency, and potential of the deep learning in delineating the individual buildings in the high-density urban village. Therefore, it should contribute to the knowledge gap-filling for remote sensing mapping of the unplanned urban settlements. This study also implies that integrating the deep learning based on U-net with the high spatial resolution satellite

images can offer accurate building information in the complex urban villages that is needed for urban redevelopment.

**Author Contributions:** Z.P. is the primary designer of this study, including data preparation, manuscript writing; J.X. and Y.G. developed the deep-learning for remote sensing data framework and implemented the programming; Y.H. gave valuable suggestions, helped with the data collection, and provided the funding for this research. G.W. secured the funding, provided critical suggestions in writing and revised the whole manuscript. All authors have read and agreed to the published version of the manuscript.

**Funding:** This research was supported by China National Key Research and Development Program (2018YFD1100801), Guangzhou Science and Technology Project (201807010048), and the International Postdoctoral Exchange Fellowship Program 2017 (Grant NO. 20170029).

**Acknowledgments:** The authors would express appreciation to the colleagues of Guangzhou Urban Renewal Bureau for their discussion.

**Conflicts of Interest:** The authors declare no conflict of interest.

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
