# Peer review of "Deep Learning Segmentation and Classification for Urban Village Using a Worldview Satellite Image Based on U-Net"

_remotesensing, doi:10.3390/rs12101574_

Round 1

Reviewer 1 Report

The paper presents UNet's application for the segmentation and classification of buildings in a densely populated and disorganized area. The paper presents a case study that is interesting because of its difficulty. The introduction is well developed, the motivation and the problem are clear. And the description of the case study and the data is very complete. However, there are some issues that need to be addressed before publication.

In abstract: "The study area located in Guangzhou City, China, with ten sites of the urban 22 villages in the central downtown area, was selected." Complete: was selectad as... or to...

"Worldview satellite image with eight pan-23 sharpened bands at a 0.5 m spatial resolution and a vector file of building boundaries were used" Complete

In introduction: "Deep learning-based semantic segmentation is to use a 101 convolutional neural network to learn prior knowledge of images and extract the features." It is a vain description

Line 185: "... to create a CNN". Not necesary, you can use a pretrained

Line 186: The numbering must follow an equal structure, the first step is a generic description, nothing is indicated to be done. The rest of the phrases have no subject, you change between present and past. Rewrite

Line 241: data augmentation needs information about anfles maximun, flips, etc

Line 245: we performed... Rewrite

Line 245: "setting hyperparameters was needed." Authors abuse these kinds of phrases. Itis not in the corresponding paragraph, and what it refers to is mentioned above. Also, hyperparameters have no to be described. Why are there different values for the same hyperparameter?

Figure 4 does not provide anything

In general there are explanations that can be eliminated: OA, recall, precision, confusion matrix, hyperparameters, are generic knowledge. A reference is enought

Sentences referencing to tables and figures must be in present: The segmentation results of the buildings ARE presented in Figure 5.

More emphasis should be placed on where the tagged data is obtained

Mention about training time and classification time of randon forest compared to UNet

Lack of comparison with other work addressing similar problems

Section 5.4 can be on analysis results and conclusions

Reviewer 2 Report

Dear authors,

Your paper on “Deep learning segmentation and classification foe urban villages using Worldview satellite image based on U-net” shows that deep-learning allows with high accuracy to map densely built-up areas, in particular, the building footprints. The paper uses a state-of-the-art deep learning architecture and makes some modification to allow using 8-band WorldView images. The mapping of building footprints is discussed in the context of information needs to support urban planning and management in China, with the specific context of urban villages. The paper is clear in the message that deep-learning is a very suitable alternative as compared to the manual delineation of building footprints. However, I have several main concerns about the paper, which would need to be addressed in a revision:

  • The introduction and the entire context of urban villages:
    • Urban sprawl is not always linked to unplanned urban development – this depends on the geographic context.
    • Unplanned urban areas – I am not certain whether urban villages are unplanned – they existed before and they were included into urban area as part of the urbanization process.
    • Unplanned urban areas – are in many cities (across the globe) a large part of the urban reality – they provide housing to poor income groups – the solution presented in the paper – the reconstruction is not necessarily a suitable solution (in particular for cities where the majority of people live in such areas – many Asian and African cities) – please refer to upgrading literature – the main issue is the improvement of housing conditions (e.g., via upgrading) and service provision.
    • In the introduction, you use various terms – unplanned – informal – slum -> without defining how they relate to urban villages.
  • The role of remote sensing – I disagree with the statement in line 54 that remote sensing can replace ground surveys – to collect detailed data on socio-economic conditions – e.g., necessary for upgrading needs ground data – while remote sensing can generate base data on the physical structures of large areas.
  • The term used in the paper “object-based classification (OBC)” is not common – please use the more standard term OBIA!
  • Line 66 -> this statement reads very strange “ not meeting the requirements of practical applications “ -> this depends and it is not clear what you mean by practical?
  • There have also been other studies mapping buildings including informal settlements e.g.,
    • https://www.mdpi.com/2306-5729/4/3/105
    •  
  • Line 149 – what is the requirement of the modern city?
  • Line 162: Please spell out the acronym NNDifuse Pan Sharpening [41] – and why is it the most optimal method?
  • Methodology – how have the parameters been selected – optimized? This should be better explained.
  • Accuracy assessment: Besides F1 score - as areas are extracted, you also need an area based accuracy assessment e.g., Jaccard index. This should be included in the assessment.
  • In the result section, you suddenly compare to OBIA and RF but this is not explained how did you use these methods – e.g., is OBIA rule-based or using RF? Later in the discussion you have the statement (line 368 ) “RF is easy to implement with less workload, but the obtained accuracy does not meet the practical application” -> this depends very much on the context and it is not clear what you mean by practical.
  • Please review and correct the references – they first reference in the list the author names are wrong - it is Wurm not Wurma!
  • The entire paper would require language editing!

Reviewer 3 Report

The research is interesting and scientific oriented. However, some suggestings are proposed for further improvement.

  1. LN156-164: I supposed that the give explanation about the source data does not seem sufficient.
  2. Figure 1: Please try to make a consistency of the legend. Why buildings (yellow color) in a separate place? Some part of the legend is embossed with the map.  
  3. Equation 1: This is not MDPI format
  4. Please add a north arrow to all maps

The orientation of the maps and figures does not clearly show the results. If you can manage the same size square for figure 5,6; it will be more readable. As a solution, try to changes the orientation of the page.   

Reviewer 4 Report

The manuscript presents a study on the use of deep learning for classifying urban villages or slums. It is an interesting study and can also be applied in estimating the population of urban villages.

The authors should consider the following in improving the manuscript.

  1. The discussion section does not properly compare the result of this study with previous research in the literature. The authors stated that previous studies did not derive building characteristics. They need to state the levels of accuracies achieved by the previous studies.
  2. In Figure 7, the result of the object-oriented classification is shown in grayscale, unlike other results that show the old buildings, new buildings and so on.
  3. The authors should consider presenting a table of classification accuracy for the RF, Object-based and U-net classifications for better comparison. 
  4. The manuscript needs extensive copy-editing. The use of English makes it difficult to fully understand the content of the article.

Round 2

Reviewer 1 Report

I congratulate the writers for all the improvements made to the article. Both the improvements and the answers are very careful and correct. I recommend its publication in its current state.

Author Response

Dear Editors and Reviewers,

       I am glad to receive the reviewer’s second-round comments, which are satisfactory to my manuscript. I sincerely appreciate all the editors and reviewers for endorsing our research. I wish you all good health and success in the future.

Here is the resubmit manuscript with a minor revision. Corresponding rectification had been highlighted in the manuscript, as well as thorough scrutiny regarding writing improvement.

Yours sincerely,

Zhuokun Pan

Postdoctoral researcher

Faner Hall, 4442, Southern Illinois University at Carbondale, IL, 62901.

Reviewer 2 Report

Dear Authors,

Thanks for the revision of your paper, which clarified and improved most of my questions and concerns. The paper is a very interesting and relevant next step to extract buildings in unplanned settlements, which is a very challenging task because of their high built-up densities.

I would have only a few very minor suggestions for a final improvement of the paper:

Line 84: Please change this sentence “the existence of unplanned settlements is regarded as a hotbed of public health and criminals” -> Please remember a large part of the urban population in Africa and Asia etc. is living in such settlement. The sentence reads very problematic stigmatizing inhabitants in such areas. Please check, e.g. the website of SDI https://knowyourcity.info/

I would suggest changing the sentence to, e.g., “Although unplanned settlements do provide housing for low-income city dwellers, their existence contributes to unequal living conditions, inhabitants often do not have access to basic services, face unhealthy living conditions and issues with public safety [4,6].” (I know most of the authors, and none would have such a statement)-> please also revisit the statement in line 349.

Line 105 “usually does not meet the requirements of effective applications” -> sentence is rather vague. Maybe change, “usually does not allow the extraction of individual buildings, but typically clusters several buildings into one segment”.

Line 120 “slums need to go beyond delineating the whole area and should characterize individual buildings.” -> depends on the purpose - but I agree that building delineation provides essential information!

Line 801 “and OBIA was implemented with ENVI 5.3.” -> please add to the text that it was a rule-based OBIA approach you used.

Line 1098 -> “This study validated the feasibility” -> please change to” This study demonstrated the feasibility”  (or showed -> you do not validate the feasibility).

Author Response

Dear Editors and Reviewers,

       I am glad to receive the reviewer’s second-round comments, which are satisfactory to my manuscript. I sincerely appreciate all the editors and reviewers for endorsing our research. I wish you all good health and success in the future.

Here is the resubmit manuscript with a minor revision. Corresponding rectification had been highlighted in the manuscript, as well as thorough scrutiny regarding writing improvement.

Yours sincerely,

Zhuokun Pan

Postdoctoral researcher

Faner Hall, 4442, Southern Illinois University at Carbondale, IL, 62901.

This manuscript is a resubmission of an earlier submission. The following is a list of the peer review reports and author responses from that submission.

Round 1

Reviewer 1 Report

In this paper, the authors present an application of the U-net model, a deep learning model, for the semantic segmentation of urban settlement in the Guangzhou city of China. While this is not really a new for the semantic segmentation of cities in the develloped word, see for example https://project.inria.fr/aerialimagelabeling/ and the paper of Huang et al 2018 cited in reference, the application in less develloped country and cities with recent very high growth is very interesting and challenging, and i feel that is the reason why this paper as the merit to be published. However, the paper need to be improve in order to publish. I have detailed my comment in the next paragraph.

Majors comments :

The authors seem to believe that the U-net model can segment individual buildings, and this is not the case. It is a model of semantic segmentation and not instance segmentation. This means that two adjacent houses won’t be separated by the U-net model, it will only give one segment. For this reason, all the comparison of house counting in the table are not informative and should be removed. Also please correct the text accordingly, to not give erroneous information to the reader.

In the paper the authors present the accuracies only with Overall accuracy and Kappa or number of houses. In Deep Learning studies a common measurement of segmentation accuracy is the F1 score (https://en.wikipedia.org/wiki/F1_score) and this is a better to explain the quality of the segmentation of the object, and it is even more important if you compare between different scenes. So i would like to see the F1-score computed for every segmentation of this paper. This will help to characterize the segmentation quality.

Please also add change in hue/saturation/brightness in the data augmentation, as you use keras there are some predefined function to do this, and you will, increase the accuracy of your model. As  you have very different reflectance for each buildings, what you can to do here is to learn to the model that the reflectance is not an important feature, this is was the data augmentation with change in hue/saturation/brightness does.

The authors also use also a lot qualitative term, such as «incredible » , « perfect » and so on, while they should give quantitative results. I understand their feeling because really the result of deep learning create this feeling, but I believe a scientific publication is not the place of such words, or it should complemented by a quantitative assessment, principally based on accuracy and numbers. So throught out the text, the authors should carefully rewrite this type of sentence with quantitative assessement.  

It is not clear how the U-net was used to do the classification of building, please add this information in the Material and Methods. May bey ou can say that you have first made with two classes, background and buiding and after with more classes to separate the buildings types old buildings, new buildings, …. 

It is not clear why you are doing this classification. If it is just a test for you, I am ok with that but it does not seems to be the case. If this is for the urban planners or for the city hall, you should emphase this the text, this is an important information.

I am not a mother tongue english speaker but i have the feeling that the paper should be corrected by a professsional english editor.

Also if the dataset of buildings can be made available along the paper on a public repository it will be very helpfull for the researcher working with such algorithm.

Minors comments :

Abstract : clarify why you make this analysis. You could make one or two sentence maximum withe 3 first sentences. Please try to synthetise a bit more and remember that is should fit in 200 worlds.  Please indicate the F1 score and Overall accuracy in the abstract

L43 : remove « negative image of a city », as this is very subjective and if you have reference for the rest of the sentence this could strenghen your point.

L122-131 explain a bit better each objective, for example «  (i) map all the buidings for ….. and then (ii) classify the segmented buildings in 5 classes cla1 cla2 cl3 in order to give more information for the urban planners.

L131 : using a deep learning method

L135 give the mean latitude/longitude of the city after ‘Guangzhou city’ 

L135 astonishing rate ? please give number and reference or remove

L139 remove ‘ugly’, too subjective

L155 give more details about the worldview image, date and nadir angle, are a minimum

L159 just a remark, in your case, the spatial information should be more important that the spectral information, i mean, all roofs seem to have different reflectance. I don’t know this particular pansharpening algorithm, but in my experience, it is better to have a pansharpen algo that conserve well the spatial information when you segment such structure, spectral info is not important. For example, the pansharpen image should look similar to the pan band and not blurry.

L164 – 165 if the polygons of building boundaries are not make directly on the image, i seriously doubt that you could make a strict alignement even with a rigourous spatial registration. As worldview does not take image at nadir there is always some distortion, which increase with building size. If you look at you image and shape closer you will see small differences. However, this is not a so important problem, and could explain why sometime the accuracies are not so high as expected. Please indicate that you have made the best possible but not that it is perfect.

L171 figure 1, please annotate as a,b,c,d,e and f . If you could increase the resoltuion i twill be reality better as we can see nothing in the WV images. Also add the background class in the legend in the last graphic (what is now transparent), and there is no need for transparency in the last graphic.

L178 remove ‘modified’

L188 change ‘superiority’ by ‘performance’

L189-199. I think you can remove this part here and fusion it with the text L208-219, which is a repetition

L200 clarify, what does mean ‘any size of image’ ?

L201 remove ‘Apparently’

L206 in the figure indicate the size in height and width of the image and filter, at least of the input and output image

L203 ’…., it can attribute a class to every single pixel.’

L223 ‘1024 samples maps’ i’m not sure of the term ‘sample’ here, please check if this is correct

L233 please also add change in hue saturation and brighness in the data augmentation. It will likely increase the accuracy of the building segmentation, as from what I see in your RGB image, such settlement have very variable reflectance values. By doing this the reflectance won’t be an important feature for the model and it will manage to find buildings with different reflectance values than the limited reflectance values present in your training sample.

L234 the learning rate is very high, may be after stabilization you could try with the same weights and a lower learning rate, this  improve a bit the model. I have saw that you use weight decay and momentum, but i prefer to only change one parameter by hand, generally it improves a bit the results.

L236 generally people use 80%  for training and 20% for validation. As you have a limited sample for training this could improve you model.

L249. Give the actual values of training and validation accuracies. You could make more data augmentation to prevent your model to overfit. Then you could run for more epochs.

L251 the speed of what ?

L250-253 please clarify, you can take other paper that use this tecnhic and see how they write the description of this graph.

L256 correct Kera by Keras

L266 this section is only accuracy assessment.

L269-271 move this to the next section, this is results

L267 – 275 this is not clear what image you use in validation during the training and if you have an independant validation with image that have not been used during the training, please clarify

L288 kappa is not an usual segment accuracy method for this type of segmentation. Please change this kappa index by the F1 score. This index give far more valuable information on your segmentaton, that are  precision, recall and the harmonic mean of them, the F1 score itself. Wikipedia as a great section on how to compute the index.

L293 please increase the resolution of the figure and and the OA and the F1 score on each predicted image. We can’t evaluate the accuracy of your segmentation just visually.

L293-306 please decribe quantitatively your results, the accuracy indices, the errors, etc… saying that it is ‘incredible’ or ‘precisely delineated’ is subjective and must not to appears in the result section.

L305 it is not an error of the Unet to merge the adjacent building, such as you have mentioned. The unet is a model for semantic segmentation and not instance segmentation soi t will never be able to separate two adjacent buildings. please correct the text. As i have said in my major comment, remove the house counting, it make no sense as unet can not separate individual buildings

L307 in the table remove number of building and add precision recall and F1 score

L309-316, here again, i can evaluate nothing visually, give the value of accuracy with Overall accuracy, precision, recall and F1 score. Please also increase the resolution of this figure 6 and add accuracy values on each predicted images. Please also describe better error.

L325 remove kappa and add precision recall and F1 score, based on this describe were it works well and where are the error for the site with less accurate results. This will help the reader to have more confidence in your results and you to understand better what are the problem of your model and how they can be resolved.

L329 -345 this is discussion not result please change the section. As i said before, Unet cannot be used for segmentation of instance so correct your text accordingly. Such paragraph, without any references or strong accuracy numbers of segmentation (such as F1 score and not overall accuracy) are too much subjective and should be avoided. For example, if my objects cover only 5% of the pixels and the model predict all wrong, can will still have 95% of overall accuracy… This is why it is better to compare with F1-score, this index will only account the the accuracy of your objects segmentation.

L346 clarify the title, for example « Performance comparison of the Unet with random forest and XXXX « 

L347 remove ‘to demonstrate superiority’ and change with something more reasonable, like « the accuracy of the Unet to segment buildings was compared to  the accuracy of XX other traditional algoritm, XX and XXX , for the same task» 

L347-363 only compare the results, don’t give any judgement here, keep this for discussion.

L365 the two first sentence could be or remove or rewrited, i cannot understand what is the point of this two sentences

L364 Please reference better your discussion (i give some example in the following comments)

L368-371 repeat the accuracy you have found, the real numbers, and also compare to the accuracies obtain by other for example on the INRIA cities dataset (Huang 2018).

L377 also, it is better to compare based on number that only by saying is better or worst, for example, you can say exactly how much the accuracy increase between the different type of classification, you gain more that 10% of accuracy in comparison to random forest. Here you can cite also the paper of alexnet in 2012, which as been the first deep learning model to beat classical agorithm, by the same amount of change in accuracy.

L382 Unet also only segment the whole build up area, you must correct this throught out your manuscrit.

L384 wht you want to say here is that to delineate each individual building in slums is not feasible with semantic segmentation model but could potentially made with instance segmentation model such Mask R CNN

L386-392 this is introduction, in discussion you should respond to this, are cnn capable of that ? what are there limit to do this ?

L401 I generally avoid this type of comments, because you can’t justified why this is not already done in your study, but your lucky i won’t ask you to do this. I think with some more data augmentation and accuracy measured with F1 score you results will be improved.

L407 this is not supported by your results, or demonstrate this in the results or remove

L441 The Conclusion is the exact repetition of you abstract. Please change it. You can here shortly repeat you main results and then write a more general conclusion such as you paragraph L416-423, which give a more general overview of the importance of deep learning for remote sensing analysis.

Reviewer 2 Report

The work uses worldview images to segment buildings via UNet. Although transfer learning between fields is welcome, the work presents serious gaps.

  • Abstract and in all article: over 80 o 90% is not a clear measurement, indicate an average accuracy
  • Line 66: "object-based" lower case
  • Line 85: "The idea of deep-learning is simple: the machine learns the features and is usually very good at decision making (classification) versus a humanity perception". Learns is very extensive. Learns to extract relevan features and classify.
  • Introduction is bad organized. Line 63 talks about ML, after, image processing, Line 82 deep learning, Line 83 semantic segmentation. Also it is not clear the link between line 82 and 83.
  • Line 102: "learning process does not require human’s prior knowledge" if you don't have to label the data
  • Line 109 (3)Also called transfer learning
  • Section 
  • Related work is missing and necessary, there are a lot of works about semantic segmentation buildings with satellite images, with or without artificial intelligence.
  • Training time is missing.
  • Las 2 paragraphs of Section 4.2 are more like a conclusion
  • The results show a remarkable overfiting that is not mentioned  or proposed a solution
  • In the discussion, a comparison with other authors is missing.
  • The conclusions should be extended

Round 2

Reviewer 1 Report

Major comments:

The Unet model is not able to separate instance that are adjacent. It is not made for this purpose. It is made from semantic segmentation. This next figure is extracted from the Figure 6 of your paper (see in the attached word document):

observed

predicted

You can see that for some of the yellow class, the buildings are not separated because when they are adjacent (For example the yellow segment in the middle on the bottom). the Unet labeled them as “building” and make only one segment. In this case how much buildings are in this segment? The Unet can’t help you to answer because it only can say ‘building’ or ‘not building’ (which is semantic segmentation); and not ‘building 1’ ‘building 2’ ‘building 3’ … ‘building N’ (which is instance segmentation). Another example is in the next figure For Garcia et al (2017) :

You model is capable of the segmentation on the left but not of the instance segmentation on the right. For example, in you case two adjacent buildings will be labelled as only one such as the two cubes on the left picture.

It is also clear in the new references you send to me for example, in the figure 13 of Zhang et al :

If you look at the column (b) second and third you will see that there are many buildings in one segment.

In your case, may be something more informative than the number of houses, which is obviously underestimated, could be the area covered by each building. I won’t ask you to do that, but maybe it could be a sufficient information to give to the urban planner to make estimates of population for example.

So, again, please check carefully and correct all sentences in your paper that make the reader believe that the Unet model is capable of instance segmentation as this is wrong.

For example, change “Automatic remote sensing mapping of single buildings in the urban village” to “Automatic remote sensing mapping of buildings in the urban village”, etc…

Garcia-Garcia, A., Orts-Escolano, S., Oprea, S., Villena-Martinez, V., & Garcia-Rodriguez, J. (2017). A review on deep learning techniques applied to semantic segmentation. arXiv preprint arXiv:1704.06857.

Zhang, P.; Ke, Y.; Zhang, Z.; Wang, M.; Li, P.; Zhang, S. Urban land use and land cover classification using novel deep learning models based on high spatial resolution satellite imagery. Sensor 2018, 18, 3717, doi:3710.3390/s18113717.

Minor comments:

Please, next time your respond to reviewers, add the line number where you have made the change, or write the change you have may in the response to reviewer document. It is very difficult to see what change you have made.

L43 : remove « negative image of a city », in the sentence “The existence of informal settlements is usually depicted as a negative image of a city, hidden with potential health risks and public unsafety [3,5].” as this is very subjective

This is not clear what image you use in validation during the training and if you have an independent validation with image that have not been used during the training, please clarify. Please indicate in the text of the paper the information you gave me in the response to reviewers (‘The program we designed has the function to partition again the training set into training and validation, during the model training process. I mean, the four sites were used for testing solely, the six sites for training are partition into 70% training and 30% validation. We had generate 10000 samples of image tiles for training process’)

The sentence in discussion on the improved performance adding more bands this is not supported by your results, please remove. You can add that in future work you will test for this.

Reviewer 2 Report

The paper has improved significantly, and the responses have been adequate

Author Response

Thank you so much for your valuable comments!